# Faraday Instability in Viscous Fluids Covered with Elastic Polymer Films

**DOI:** 10.3390/polym14122334

**Published:** 2022-06-09

**Authors:** Junxiu Liu, Wenqiang Song, Gan Ma, Kai Li

**Affiliations:** 1Anhui Province Key Laboratory of Building Structure and Underground Engineering, Anhui Jianzhu University, Hefei 230601, China; tjuliu@ahjzu.edu.cn; 2College of Civil Engineering, Anhui Jianzhu University, Hefei 230601, China; swq20222022@163.com (W.S.); mg201820212022@163.com (G.M.)

**Keywords:** Faraday instability, viscous fluid, elastic polymer film, stability criterion, Floquet theory

## Abstract

Faraday instability has great application value in the fields of controlling polymer processing, micromolding colloidal lattices on structured suspensions, organizing particle layers, and conducting cell culture. To regulate Faraday instability, in this article, we attempt to introduce an elastic polymer film covering the surface of a viscous fluid layer and theoretically study the behaviors of the Faraday instability phenomenon and the effect of the elastic polymer film. Based on hydrodynamic theory, the Floquet theory is utilized to formulate its stability criterion, and the critical acceleration amplitude and critical wave number are calculated numerically. The results show that the critical acceleration amplitude for Faraday instability increases with three increasing bending stiffness of the elastic polymer film, and the critical wave number decreases with increasing bending stiffness. In addition, surface tension and viscosity also have important effects on the critical acceleration amplitude and critical wave number. The strategy of controlling Faraday instability by covering an elastic polymer film proposed in this paper has great application potential in new photonic devices, metamaterials, alternative energy, biology, and other fields.

## 1. Introduction

In 1831, Faraday observed that when a container containing liquid is placed on the base of periodic vertical oscillation, a standing wave can be observed on the surface of the liquid [1]. If the forced amplitude is greater than the critical threshold, the excitation will lead to the formation of subharmonic with a response frequency of half of the excitation frequency. Later, experiments by Lord Rayleigh in 1883 verified Faraday’s observational conjecture [2]. The analytic solution of the linear stability analysis for ideal fluids was provided by Benjamin and Ursell in 1954 [3]. Then, Rajchenbach and Clamond defined the trigger of Faraday instability with critical driving force [4], which has been prescribed as a hydrodynamic regime for ordering surface waves in the macroscopic realm [5,6].

Faraday instability has a wide range of applications in the field of fluid mechanics and other fields. For example, Faraday instability can be used to make measurement equipment or measure material parameters, such as measuring the surface tension of soft materials [7], developing new photonic devices [8,9], metamaterials [10,11], etc. In addition, it has potential applications in biology [12] and medicine, such as application in cell culture patterns [13,14] and detecting physiological processes of organisms [15]. It can also be used to monitor and control animal behavior such as earthworms [16], sense and modify soil structure as well as to increase crop yields [17,18,19], medical examinations of medical ultrasound and photoacoustic imaging modalities [20], Brillouin Light Scattering spectroscopy [21], laser vibrometry [22] and develop new methods of eradicating viruses and bacteria [23,24,25].

In recent years, people have carried out in-depth research and exploration of Faraday instability. For example, Lyubimova et al. studied Faraday waves on band patterns under zero gravity conditions [26]. Lioubashevski et al. conducted Faraday instability studies on highly dissipative fluids [27,28]. In response to the Faraday instability phenomenon of viscous fluids, Kumar proposed a quantitative linear theory for viscous liquids and concluded that subharmonic resonance dominates in the case of smaller viscosity; due to the existence of a rigid substrate at the bottom, the viscosity effect is enhanced, and the double critical points of two modes may be encountered in low height containers [29]. Further, Kumar and Bajaj conducted a linear stability analysis by considering the viscous boundary conditions of the interface between two viscous fluids and concluded that the advantages of the resonant mode over other modes were mainly controlled by the ratio between the height of the container and the length of the viscous boundary on the free surface [30,31]. Christiansen et al. further found that the existence of unbounded patterns of nonlinear surface waves at a large aspect ratio has been reported in highly viscous fluids close to the Faraday instability [32,33,34,35], where wave ordering is scaffolded upon bulk frictional stresses [35,36]. Raynal et al. conducted experiments on non-Newtonian liquids and observed that the instability threshold increases with the increase in driving frequency, indicating that the viscosity of the solution will decrease instead [37]. For the study of the Faraday instability of inviscid bounded fluids, container-bounded Faraday waves have been employed to create standing patterns in inviscid liquids [11,38], in analogy with the parametric excitations of quantum condensates [39,40] and the optical lattices able to confine ultra-cold atom gases [41].

In addition, the Faraday instability phenomenon has been extended to elastic bodies as well as living organisms. Müller et al. studied linear viscoelastic fluids and found that the resonance modes become harmonics in the range where the elastic force is in the same order of magnitude as the surface tension at the free boundary [34,42]. Giulia et al. conducted an experimental analysis of soft elastic bodies and used Floquet theory to describe the Faraday wave in soft elastic bodies. It was found that the Faraday wave in soft elastic bodies was dominated by harmonic resonance modes, which provided a method for distinguishing fluids from soft elastic bodies [43]. Maksymov and Pototsky excited Faraday wave in living earthworms, which can be used to detect and possibly control important physiological processes of organisms, providing a method for solving fundamental biological problems, and developing new technology for detecting and controlling biological and physiological processes in vivo [15].

The regulation of Faraday instability is very crucial for its applications. In recent decades of research, people have explored some methods to regulate Faraday instability, such as adjusting the viscosity of viscous fluid [30] and controlling boundary conditions [44,45]. In addition, Douady and Fauve conducted an experimental study of parametrically excited surface wave modes and found that the pattern wave number is fixed by the excitation frequency, whereas the shape depends on the nonlinear interaction [46]. Wright and Saylor found that Faraday waves can be used to generate linear particle patterns with controllable length scales on glass substrates [47]. Mahr and Rehberg found that parametrically driven surface waves in ferrofluids can be excited by an external oscillating magnetic field, and the static magnetic field changes the restoring force and damping coefficients of various surface waves [48]. Samanta performed Faraday instability experiments on viscous liquids on the plane of the porous layer, and found that the critical amplitude for Faraday instability decreases with increasing permeability values, and the presence of the porous layer leads to a faster transition process from subharmonic instability to harmonic instability in the wavenumber regime [49]. In addition, the conditions under which Faraday instability of the surfactant-coated liquid layer occurs were investigated [50,51]. Kharbedia et al. changed the stiffness of the fluid surface by adsorbing a rigid film of soluble surfactant on the fluid surface and then found that the stiffness of the liquid surface has an important influence on the Faraday wave formation [52]. Recently, in the field of solution of Janus colloidal particles in a solvent, by tuning the magnitude of anisotropic colloid–solvent interactions, Faraday waves of different amplitudes are generated [53,54]. 

Inspired by the above works, we attempt to introduce an elastic polymer film covering viscous fluids to control its Faraday instability. Through theoretical research, we obtained the critical criterion of Faraday instability of the fluid surface covered with an elastic polymer film. We also investigate the influence of the elastic polymer film on the critical acceleration amplitude and the critical wave number of the Faraday instability so as to provide guidance for its applications. Compared to existing methods such as changing the viscosity, this strategy is simple to operate and cost-effective. The text of this paper is structured as follows. Firstly, we establish the theoretical model of a viscous fluid layer covered with an elastic polymer film and deduce the surface Faraday instability criterion by using Floquet theory in Section 2. Secondly, we use MATLAB software to numerically calculate the critical acceleration amplitude and the critical wave number. Then we discuss the influence of the bending stiffness of the elastic polymer covering layer on the Faraday instability in Section 3. Finally, we draw conclusions based on the numerical analysis results. 

## 2. Model and Formulation

In this section, we first establish the theoretical model of a viscous fluid layer covered with an elastic polymer film and then deduce the stability criterion of surface Faraday instability by using Floquet theory. Finally, the solution method for the critical magnitude and critical wavenumber is presented. 

### 2.1. Governing Equations

Figure 1 shows an incompressible viscous fluid layer covered with an elastic polymer film, which is placed on a rigid substrate. We assume that the thickness H of fluid layer is large enough so that the effects of the structure of substrate and the magnitude of fluid–substrate and polymer–substrate interactions on the Faraday wave amplitude are negligible. We assume that the density ρ and dynamic viscosity ν of fluid are constant, and the viscosity of air is not considered. The system is affected by the vertical sinusoidal acceleration of amplitude ε and frequency ω, which can drive the viscous fluid layer to move and lead to the formation of standing waves. In the reference frame fixed with the vibrating rigid substrate, the fluid dynamics are controlled by Navier–Stokes equation
(1)$$\rho \left(\partial_{t}u+u\cdot \nabla u\right)=-\nabla p+\nu \nabla^{2}u+\rho G\left(t\right)k,$$
and continuity equation
(2)∇⋅u=0,
where, u is the velocity, and G(t) is the equivalent gravitational acceleration, i.e.,
(3)G(t)=g−εcos(ωt).

Taking the curl of Equation (1) twice, we can obtain
(4)$$\left(\partial_{t}-\mu \nabla^{2}\right)\nabla^{2}w=0,$$
where the kinematic viscosity of the fluid μ=ν/ρ. 

Next, we consider the boundary condition on the rigid plate. Since the fluid rests on the rigid plate, the velocity field should be zero. Thus, we have
(5)$$w|{}_{z=-H}=0,$$
and
(6)$$\partial_{z}w|{}_{z=-H}=0.$$

Free surface boundary conditions are further considered. The free surface is initially horizontally stationary and lies in the plane of the coordinate at z=0. When the rigid substrate vibrates, the free surface is located at z=h(x,y,t), and subjects to kinematic boundary condition [55],
(7)$${\left[\partial_{t}+(u\cdot \nabla )\right]h=w|}_{z=h}.$$

Through linearization, the kinematic condition Equation (7) can be simplified as
(8)$${\partial_{t}h=w|}_{z=0}.$$

In addition, the stress tensor on the free surface can be written as
(9)$$\tau_{jk}=-p\delta_{jk}+\nu (\partial_{j}u_{k}+\partial_{k}u_{j})+\rho G(t)h\delta_{jz}\delta_{kz},$$
where the last term is the stress due to surface deformation under the equivalent gravitational acceleration. We assume that there are no tangential stress components on the free surface, i.e., $\tau_{xz}=\tau_{yz}=0$. Since these stress components disappear at the free surface, we can obtain that
(10)$$\nu (\partial_{x}\tau_{xz})|{}_{z=0}=\nu (\partial_{y}\tau_{yz})|{}_{z=0}=0.$$

By substituting boundary condition Equation (10) into Equation (9) and using continuity Equation (2), we can obtain
(11)$$\nu (\nabla_{H}^{2}-\partial_{ZZ})w|{}_{z=0}=0.$$

Meanwhile, the normal component of the stress tensor on the free surface originates from the contributions of both the surface tension and bending of elastic polymer film. For the linear case at small curvatures, the contribution of surface tension is the free surface tension γ times the curvature of the free surface. The contribution of elastic film bending can be provided by the effective lateral pressure, which is $B\nabla_{H}^{4}h$ with bending stiffness *B* of the film [56]. Therefore, we can obtain
(12)$$\tau_{zz}|{}_{z=0}=\gamma \nabla_{H}^{2}h-B\nabla_{H}^{4}h.$$

By substituting Equation (9) of the stress tensor into Equation (12), the expression for the normal pressure at free surface is obtained,
(13)$$p|{}_{z=h}=2\nu {\left(\partial_{z}w\right)}_{z=0}+\rho G\left(t\right)h-\gamma \nabla_{H}^{2}h+B\nabla_{H}^{4}h.$$

Taking the horizontal divergence of Equations (1) and applying Equation (2), another expression for pressure can be derived:(14)$$\nabla_{H}^{2}p=(\nu \nabla^{2}-\rho \partial_{t})\nabla_{H}\cdot u_{H}=(\rho \partial_{t}-\nu \nabla^{2})\partial_{z}w.$$

By eliminating p in Equations (13) and (14), we can obtain
(15)$${\left[\left(\rho \partial_{t}-\nu \nabla^{2}\right)\partial_{z}w\right]}_{z=0}=2\nu {\left(\nabla_{H}^{2}\partial_{z}w\right)}_{z=0}+\rho G\left(t\right)\nabla_{H}^{2}h-\gamma \nabla_{H}^{4}h+B\nabla_{H}^{6}h,$$
at the free surface, which is an additional boundary condition for Equation (4) and is the only equation for which the external forcing G(t) appears explicitly.

The dynamic Equation (4), accompanied by boundary conditions (5), (6), (8), (11), and (15), governs the Faraday instability of the fluid layer covered with the elastic polymer film under the periodic excitation of the rigid substrate. The stability analysis will be carried out below.

### 2.2. Stability Analysis

To perform stability analysis, we first express the velocity u and surface deformation h(x,y,t) in the normal modes $\sin(k_{x}x+k_{y}y)$ with the horizontal wave number $k=\sqrt{k_{x}^{2}+k_{y}^{2}}$. We now replace w(x,z,t) with $w(z,t)\sin(k_{x}x+k_{y}y)$, h(x,y,t) with $h(t)\sin(k_{x}x+k_{y}y)$ and the differential operator $\nabla_{H}^{2}$ with the number $-k^{2}$. By introducing the following dimensionless parameters: $\tilde{\varepsilon }=\frac{\varepsilon }{g}$, $\tilde{\omega }=\omega \sqrt{\frac{H}{g}}$, $\tilde{\upsilon }=\frac{\upsilon }{\rho \sqrt{gH^{3}}}$, $\tilde{\mu }=\frac{\mu }{\sqrt{gH^{3}}}$, $\tilde{k}=kH$, $\tilde{w}=\frac{w}{\sqrt{gH}}$, $\tilde{h}=\frac{h}{H}$, $\tilde{s}=s\sqrt{\frac{H}{g}}$, $\tilde{t}=t\sqrt{\frac{g}{H}}$, $\tilde{z}=\frac{z}{H}$, $\tilde{\gamma }=\frac{\gamma }{\rho gH^{2}}$, $\tilde{B}=\frac{B}{\rho gH^{4}}$, ${\tilde{\kappa }}_{n}=\kappa_{n}H$, and $\tilde{\delta }=\delta \sqrt{\frac{H}{g}}$, Equations (4)–(6), (8), (11) and (15) can be further rewritten as
(16)$$\left[\partial_{\tilde{t}}-\tilde{\mu }(\partial_{\tilde{z}\tilde{z}}-{\tilde{k}}^{2})\right](\partial_{\tilde{z}\tilde{z}}-{\tilde{k}}^{2})\tilde{w}=0,$$
(17)$${\left(\partial_{\tilde{z}\tilde{z}}\tilde{w}+{\tilde{k}}^{2}\tilde{w}\right)}_{\tilde{z}=0}=0,$$
(18)$${\tilde{w}|}_{\tilde{z}=-H}=0,$$
(19)$${\left(\partial_{\tilde{z}}\tilde{w}\right)}_{\tilde{z}=-H}=0,$$
(20)$${\left[\left(\partial_{\tilde{t}}-\tilde{\nu }\partial_{\tilde{z}\tilde{z}}+3\tilde{\nu }{\tilde{k}}^{2}\right)\partial_{\tilde{z}}\tilde{w}\right]}_{\tilde{z}=0}=-\left[\tilde{G}(\tilde{t})+\tilde{\gamma }{\tilde{k}}^{2}+\tilde{B}{\tilde{k}}^{4}\right]{\tilde{k}}^{2}\tilde{h},$$
(21)$$\partial_{\tilde{t}}\tilde{h}={\tilde{w}|}_{\tilde{z}=0}.$$

The above set of Equations (16)–(21) describe the complete linear stability problem for a viscous fluid layer covered with an elastic polymer film under parametric oscillations.

Next, we apply the Floquet theory to analyze the stability problems of Equations (16)–(21). Considering that the equivalent gravitational acceleration $\tilde{G}\left(\tilde{t}\right)$ in the vibrating coordinate system is a periodic function, the solutions of Equations (16)–(21) are assumed to be in Floquet form [30]. Then, the surface deformation $\tilde{h}$ can be expressed as:(22)$$\tilde{h}(\tilde{t})=e^{\tilde{\delta }\tilde{t}}\sum_{n=-\infty }^{\infty }{\tilde{h}}_{n}e^{in\tilde{\omega }\tilde{t}},$$
where, the Floquet exponent $\tilde{\delta }$ can be expressed as
(23)$$\tilde{\delta }=\tilde{s}+i\alpha \tilde{\omega },$$
where $\tilde{s}$ and α can take any real and finite values. The solutions $\tilde{h}(\tilde{t})$ corresponding to α=0 and α=1/2 are named as harmonic and subharmonic solutions, respectively.

Similarly, we assume that $\tilde{\omega }\left(\tilde{z},\tilde{t}\right)$ can be expressed as:(24)$$\tilde{w}(\tilde{z},\tilde{t})=e^{\tilde{\delta }\tilde{t}}\sum_{n=-\infty }^{\infty }{\tilde{w}}_{n}(\tilde{z})e^{in\tilde{\omega }\tilde{t}}.$$

Substituting Equation (24) into Equation (16), we obtain the following Fourth-order ordinary differential equation in $\tilde{z}$:(25)$$(\partial_{\tilde{z}\tilde{z}}-{\tilde{k}}^{2})(\partial_{\tilde{z}\tilde{z}}-{\tilde{\kappa }}_{n}^{2}){\tilde{w}}_{n}(\tilde{z})=0,$$
where,
(26)$${\tilde{\kappa }}_{n}^{2}\equiv {\tilde{k}}^{2}+\frac{\left[\tilde{s}+i(\alpha +n)\tilde{\omega }\right]}{\tilde{\mu }}.$$

In computing the ${\tilde{\kappa }}_{n}$, we should use the root of Equation (26) having positive real part.

The general solution of Equation (25) is
(27)$${\tilde{w}}_{n}(z)={\tilde{F}}_{n}\cosh(\tilde{k}\tilde{z})+{\tilde{E}}_{n}\sinh(\tilde{k}\tilde{z})+{\tilde{L}}_{n}\cosh({\tilde{\kappa }}_{n}\tilde{z})+{\tilde{K}}_{n}\sinh({\tilde{\kappa }}_{n}\tilde{z}).$$

Inserting Equation (27) in Equations (17)–(21) and using Equation (27) lead to
(28)$${\tilde{F}}_{n}=\tilde{\mu }({\tilde{\kappa }}_{n}^{2}+{\tilde{k}}^{2}){\tilde{h}}_{n,}$$
(29)$${\tilde{L}}_{n}=-2\tilde{\mu }{\tilde{k}}^{2}{\tilde{h}}_{n},$$
(30)$${\tilde{K}}_{n}=-\frac{\left[\tilde{k}F_{n}+{\tilde{L}}_{n}\{\tilde{k}\cosh({\tilde{\kappa }}_{n})\cosh(\tilde{k})-{\tilde{\kappa }}_{n}\sinh({\tilde{\kappa }}_{n})\sinh(\tilde{k})\}\right]}{\left[{\tilde{\kappa }}_{n}\cosh({\tilde{\kappa }}_{n})\sinh(\tilde{k})-\tilde{k}\sinh({\tilde{\kappa }}_{n})\cosh(\tilde{k})\right]},$$
(31)$${\tilde{E}}_{n}=\frac{1}{\tilde{k}}\left\{\begin{array}{l} {\tilde{L}}_{n}\left[{\tilde{\kappa }}_{n}\sinh({\tilde{\kappa }}_{n})\cosh\left(\tilde{k}\right)-\tilde{k}\cosh\left({\tilde{\kappa }}_{n}\right)\sinh\left(\tilde{k}\right)\right]\\ -{\tilde{K}}_{n}\left[{\tilde{\kappa }}_{n}\cosh\left({\tilde{\kappa }}_{n}\right)\cosh\left(\tilde{k}\right)-\tilde{k}\sinh\left({\tilde{\kappa }}_{n}\right)\sinh\left(\tilde{k}\right)\right] \end{array}\right\}.$$

Plugging Equation (11) into Equation (21) and using Equations (27)–(31), we obtain the following recurrence relation:(32)$${\tilde{A}}_{n}{\tilde{h}}_{n}=\tilde{\varepsilon }({\tilde{h}}_{n-1}+{\tilde{h}}_{n+1}),$$
where,
(33)$${\tilde{A}}_{n}=2+2\tilde{\gamma }{\tilde{k}}^{2}+2\tilde{B}{\tilde{k}}^{4}-2{\tilde{\mu }}^{2}\frac{4{\tilde{\kappa }}_{n}{\tilde{k}}^{2}\left({\tilde{\kappa }}_{n}{}^{2}+{\tilde{k}}^{2}\right)-{\tilde{S}}_{n}\cosh{\tilde{\kappa }}_{n}\cosh\tilde{k}+{\tilde{T}}_{n}\sinh{\tilde{\kappa }}_{n}\sinh\tilde{k}}{\tilde{k}{\tilde{\kappa }}_{n}\cosh{\tilde{\kappa }}_{n}\sinh\tilde{k}-{\tilde{k}}^{2}\sinh{\tilde{\kappa }}_{n}\cosh\tilde{k}},$$

With
(34)$${\tilde{S}}_{n}={\tilde{\kappa }}_{n}\left({\tilde{\kappa }}_{n}{}^{4}+2{\tilde{\kappa }}_{n}{}^{2}{\tilde{k}}^{2}+5{\tilde{k}}^{4}\right),$$
(35)$${\tilde{T}}_{n}=\tilde{k}\left({\tilde{\kappa }}_{n}{}^{4}+6{\tilde{\kappa }}_{n}{}^{2}{\tilde{k}}^{2}+{\tilde{k}}^{4}\right).$$

When $\tilde{\delta }+in\tilde{\omega }=0$, ${\tilde{\omega }}_{0}(\tilde{z})=0$ for all $\tilde{z}$, and we can obtain that
(36)$${\tilde{A}}_{0}(\tilde{\delta }+in\tilde{\omega }=0)\equiv {\tilde{A}}_{0}^{H}{\tilde{A}}_{0}^{H}=2+2\tilde{\gamma }{\tilde{k}}^{2}+2\tilde{Β}{\tilde{k}}^{4}.$$

### 2.3. Solution

Generally, the external forced coupling of different temporal modes makes accurate stability analysis of arbitrary viscosities difficult. Here, to determine the stability of free surface of the viscous fluid layer, we convert the recurrence relation Equation (32) into a matrix equation $\tilde{M}\tilde{h}=\tilde{\varepsilon }\tilde{N}\tilde{h}$, where $\tilde{M}$ is a diagonal square matrix with complex elements, and $\tilde{N}$ is a banded square matrix with two sub-diagonals. Further, a common eigenvalue problem can be easily constructed by inverting $\tilde{M}$:(37)$$({\tilde{M}}^{-1}\tilde{N})\tilde{h}=\frac{1}{\tilde{\varepsilon }}\tilde{h}.$$

For the case of harmonic, the matrix ${\tilde{M}}^{-1}\tilde{N}$ reads
(38)$${\tilde{M}}^{-1}\tilde{N}=\left(\begin{array}{ccccccc} \vdots & \vdots & \vdots & \vdots & \vdots & \vdots & \vdots \\ \cdots & 0 & 1/{\tilde{A}}_{2}^{*} & 0 & 0 & 0 & \cdots \\ \cdots & 1/{\tilde{A}}_{1}^{*} & 0 & 1/{\tilde{A}}_{1}^{*} & 0 & 0 & \cdots \\ \cdots & 0 & 1/{\tilde{A}}_{0}^{H} & 0 & 1/{\tilde{A}}_{0}^{H} & 0 & \cdots \\ \cdots & 0 & 0 & 1/{\tilde{A}}_{1}^{} & 0 & 1/{\tilde{A}}_{1}^{} & \cdots \\ \cdots & 0 & 0 & 0 & 1/{\tilde{A}}_{2}^{} & 0 & \cdots \\ \vdots & \vdots & \vdots & \vdots & \vdots & \vdots & \vdots \end{array}\right).$$

For the case of subharmonic, the matrix ${\tilde{M}}^{-1}\tilde{N}$ is shown as
(39)M˜−1N˜=(⋮⋮⋮⋮⋮⋮⋯01A˜1*00⋯⋯1A˜0*01A˜0*0⋯⋯01A˜001A˜0⋯⋯001A˜10⋯⋮⋮⋮⋮⋮⋮).

For harmonic and subharmonic cases, we can obtain the eigenvalue of matrix ${\tilde{M}}^{-1}\tilde{N}$ by numerical calculation. Then, one can obtain the tongue-like neutral stability curves in the $\tilde{\varepsilon }\text{-}\tilde{k}$ plane at a provided driving frequency. From the curve with the lowest acceleration amplitude, the stability threshold and critical wave number can be predicted for a provided set of problem parameters. 

## 3. Results and Discussion

In this section, we first use MATLAB software to numerically calculate the critical acceleration amplitude and the critical wave number for the Faraday instability and then discuss the influence of the bending stiffness of the elastic polymer covering layer on the Faraday instability. 

### 3.1. Stability Criterion

The typical values of material properties and geometric parameters from accessible experiments [29,30,51] are listed in Table 1, and the estimated values of dimensionless parameters are listed in Table 2. Figure 2a,b draw the $\tilde{\varepsilon }\text{-}\tilde{k}$ marginal stability curves of the subharmonic mode and harmonic mode of Faraday instability of viscous fluids covered with an elastic polymer film. In the calculation, we set the dimensionless parameters of driving frequency as $\tilde{\omega }=5.33$, surface tension as $\tilde{\gamma }=1.7$, and kinematic viscosity as $\tilde{\mu }=0.35$. As can be seen from Figure 2, the tongue-shaped curve of subharmonic (SH) mode and harmonic (H) mode appears alternately, and the three tongue-shaped curves all have the lowest point. This result is similar to the previous Faraday instability result of the fluid without the elastic polymer film [29], and the viscous fluid covered with the elastic polymer film can still trigger Faraday instability. Moreover, the lowest point increases successively; that is, under the provided driving frequency $\tilde{\omega }$ and surface tension $\tilde{\gamma }$, the lowest point of SH mode is the smallest, which means that the Faraday instability of subharmonic mode always occurs first in the viscous fluid. The acceleration amplitude $\tilde{\varepsilon }$ corresponding to the smallest SH mode is the critical acceleration amplitude ${\tilde{\varepsilon }}_{c}$ of Faraday instability on the fluid surface, and the corresponding critical wave number ${\tilde{k}}_{c}$ is the wave number of Faraday instability surface morphology. The result means that when the amplitude $\tilde{\varepsilon }$ reaches ${\tilde{\varepsilon }}_{c}$, the Faraday instability of the fluid surface will be triggered, and the wave number of the instability morphology is ${\tilde{k}}_{c}$. If we further increase the acceleration amplitude, Faraday instability in other modes may be triggered. The Faraday instability presented here can be degraded to the case of viscous fluids without elastic polymer film, as shown in Figure 2b for the case of $\tilde{B}=0$, and the results are consistent with those in Kumar’s paper [29]. 

Based on the critical amplitude ${\tilde{\varepsilon }}_{c}$ and the critical wavenumber ${\tilde{k}}_{c}$ obtained from the marginal stability curve $\tilde{\varepsilon }\text{-}\tilde{k}$ in Figure 2, we further discuss the relationship between the critical amplitude ${\tilde{\varepsilon }}_{c}$ and the driving frequency $\tilde{\omega }$ when Faraday instability occurs. Figure 3 shows the variation of the critical amplitude ${\tilde{\varepsilon }}_{c}$ of SH mode and H mode with the driving frequency $\tilde{\omega }$ in the Faraday instability of the viscous fluid covered with the elastic polymer film. We set the dimensionless parameter of surface tension as $\tilde{\gamma }=1.7$ and kinematic viscosity as $\tilde{\mu }=0.35$. As can be seen from Figure 3, the critical amplitude ${\tilde{\varepsilon }}_{c}$ increases as the driving frequency $\tilde{\omega }$ increases for both H and SH modes. This result implies that larger acceleration amplitudes are required to trigger Faraday instability of the viscous fluid at higher driving frequencies. This is because when the driving frequency $\tilde{\omega }$ becomes larger, the viscous fluid generates more resistance and energy consumption, which makes it difficult to trigger Faraday instability. This influence trend is different from that of soft elastomers. In the case of soft elastomer layers, the larger the driving frequency, the smaller the critical amplitude [43]. This result shows that people can increase the driving frequency to prevent Faraday instability of viscous fluid from causing safety hazards. In addition, we note that the critical amplitude of Faraday instability in SH mode is always lower than that in H mode. This shows that under the same driving frequency and surface tension, the critical amplitude of viscous fluid instability covered with the elastic polymer elastic layer always corresponds to the critical value of SH resonance.

Figure 4 shows the variation of the critical wave number ${\tilde{k}}_{c}$ of SH mode and H mode with the driving frequency $\tilde{\omega }$ in the Faraday instability of the viscous fluid covered with the elastic polymer film. We set the dimensionless parameter of surface tension as $\tilde{\gamma }=1.7$ and kinematic viscosity as $\tilde{\mu }=0.35$. As can be seen from Figure 4, the critical wave number ${\tilde{k}}_{c}$ increases with the increase in driving frequency $\tilde{\omega }$ for both H and SH modes. The results show that when the driving frequency $\tilde{\omega }$ increases, the wave number caused by Faraday instability of the viscous fluid will increase when the surface tension is provided. This is because the higher the driving frequency, the smaller the wavelength destabilization patterns can appear on the surface of the viscous fluid. This result means that we can adjust the Faraday instability morphology by changing the frequency to meet the needs of target applications. In addition, the critical wave number of Faraday instability in H mode is always higher than that in SH mode; that is, under the same frequency and surface tension, the instability morphology of fluid surface always occurs at the instability wavelength of SH mode.

### 3.2. Effect of Surface Tension and Bending Stiffness

For provided material and structure of the polymer film, its surface tension and bending stiffness are usually interrelated [57]. In this work, we discuss the effects of surface tension and bending stiffness on the critical value of Faraday instability, respectively, to understand the effect of each parameter on Faraday instability. The effect of surface tension $\tilde{\gamma }$ on Faraday instability is investigated next. Figure 5 shows the variation of the critical amplitude ${\tilde{\varepsilon }}_{c}$ of SH mode and H mode with the surface tension $\tilde{\gamma }$ in the Faraday instability of the viscous fluid covered by the elastic polymer film. Here, we set the dimensionless parameter of the drive frequency as $\tilde{\omega }=5.33$ and kinematic viscosity as $\tilde{\mu }=0.35$ during the calculation. As can be seen from Figure 5, the critical amplitude ${\tilde{\varepsilon }}_{c}$ increases with the increase in surface tension $\tilde{\gamma }$ for both H and SH modes. This result shows that the larger the surface tension, the larger the amplitude required to trigger the Faraday instability of the viscous fluid. This is because when the surface tension increases, the viscous fluid consumes more surface energy when it is unstable, resulting in difficulty in instability, which is in line with physical intuition. This result means that we can control the occurrence conditions of Faraday instability by changing the surface tension to meet the application requirements. Similarly, the critical amplitude of Faraday instability in H mode is always higher than that in SH mode; that is, under the same surface tension, the critical amplitude of viscous fluid instability covered with the elastic polymer film always corresponds to the critical amplitude of the SH mode.

Figure 6 shows the variation of the critical wave number ${\tilde{k}}_{c}$ of SH mode and H mode with the surface tension $\tilde{\gamma }$ in the Faraday instability of the viscous fluid covered with the elastic polymer film. During the calculation, we set the dimensionless parameter of the drive frequency as $\tilde{\omega }=5.33$ and kinematic viscosity as $\tilde{\mu }=0.35$. As a result, it can be seen from Figure 6 that the critical wave number ${\tilde{k}}_{c}$ decreases as the surface tension $\tilde{\gamma }$ increases for both H and SH modes. The results show that when the surface tension $\tilde{\gamma }$ increases, the wave number ${\tilde{k}}_{c}$ of Faraday instability of the viscous fluid will decrease. That is because when the surface tension increases, the surface energy of the viscous fluid surface becomes larger when it is unstable, resulting in a larger wavelength, and thus the wave number will decrease accordingly. In addition, the critical wave number of Faraday instability of the viscous fluid in H mode is higher than that in SH mode, indicating that Faraday instability always occurs first in SH mode under the same surface tension.

Next, we explore the influence of the bending stiffness $\tilde{B}$ of the elastic polymer film on the critical amplitude ${\tilde{\varepsilon }}_{c}$ and critical wave number ${\tilde{k}}_{c}$ of the Faraday instability of viscous fluid. In Figure 7, we draw the variation of the critical amplitude ${\tilde{\varepsilon }}_{c}$ of Faraday instability with the bending stiffness $\tilde{B}$ of the elastic polymer film. In the calculation process, we set dimensionless parameters of driving frequency as $\tilde{\omega }=5.33$, surface tension as $\tilde{\gamma }=1.7$, and kinematic viscosity as $\tilde{\mu }=0.35$. As can be seen from Figure 7, the critical amplitude ${\tilde{\varepsilon }}_{c}$ increases with the increase in bending stiffness $\tilde{B}$ for both H and SH modes. The results show that when the bending stiffness increases, the critical amplitude of Faraday instability on the viscous fluid surface covered with the elastic polymer film will also increase. This is because when the elastic polymer film becomes thicker, the viscous fluid will become more difficult to destabilize under the same provided parameters, and thus the critical amplitude ${\tilde{\varepsilon }}_{c}$ will also increase. That is, it has a stabilizing effect when the bending stiffness becomes larger. Therefore, we can control the occurrence of Faraday instability phenomenon by covering the surface of the viscous fluid layer with an elastic polymer film. In this way, the harm of Faraday instability can be avoided, and the Faraday instability phenomenon can be exploited by using this regulation method.

Figure 8 shows the variation of the critical wave number ${\tilde{k}}_{c}$ of Faraday instability on the viscous fluid surface covered with the elastic polymer film with bending stiffness $\tilde{B}$. Here, we set dimensionless parameters of driving frequency as $\tilde{\omega }=5.33$, surface tension as $\tilde{\gamma }=1.7$, and kinematic viscosity as $\tilde{\mu }=0.35$. As can be seen from Figure 8, the critical wave number ${\tilde{k}}_{c}$ decreases with the increase in bending stiffness $\tilde{B}$ for both H and SH modes. The results show that when the bending stiffness $\tilde{B}$ increases, the critical wave number ${\tilde{k}}_{c}$ of Faraday instability will decrease. This result means that when the elastic polymer film becomes thicker, the viscous fluid surface is more difficult to vibrate and stand up. When the viscous fluid surface is unstable, the surface energy is greater, resulting in an increase in wavelength, and thus the wave number will be reduced. Therefore, we can adjust the Faraday instability morphology by changing the bending stiffness of the elastic polymer film to meet the needs of target applications.

## 4. Conclusions

Faraday instability has great application value in the fields of polymer processing, micromolding colloidal lattice on structured suspension, tissue and granular layer, and cell culture. In this paper, we introduce an elastic polymer film covering a viscous fluid layer and attempt to precisely and easily control the Faraday instability phenomenon of the viscous fluid layer. Based on Floquet theory, we theoretically investigate the surface Faraday instability of a viscous fluid layer covered with an elastic polymer film. The numerical calculations show that the critical acceleration amplitude of Faraday instability increases with the increasing bending stiffness of the film, and the critical wave number decreases with the increase in the bending stiffness. In addition, the driving frequency and surface tension also have some effects on the critical acceleration amplitude and critical wave number. These variation trends are the same for both H and SH modes, for these parameters always inhibit or promote the stability of the system. Compared with existing methods such as changing the viscosity of the fluid, our strategy of covering an elastic polymer film is simple to operate and cost-effective. The regulation methods proposed in this paper also have great application potential in new photonic devices, metamaterials, alternative energy, and the field of biology. 

## Figures and Tables

**Figure 1 polymers-14-02334-f001:**
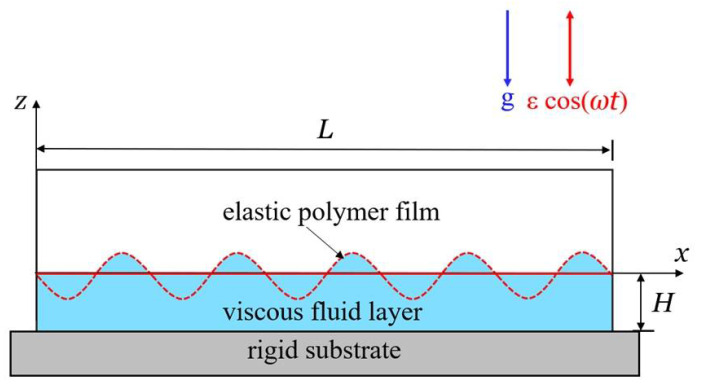
Schematic illustration of Faraday instability of a viscous fluid layer covered with an elastic polymer film. The length of container is L, and the thickness of viscous fluid layer is H. The bending stiffness and thickness of the elastic polymer film are B and t, respectively. The container is placed on a rigid substrate, and subjected to its own weight and a vertical sinusoidal acceleration with amplitude ε and frequency ω.

**Figure 2 polymers-14-02334-f002:**
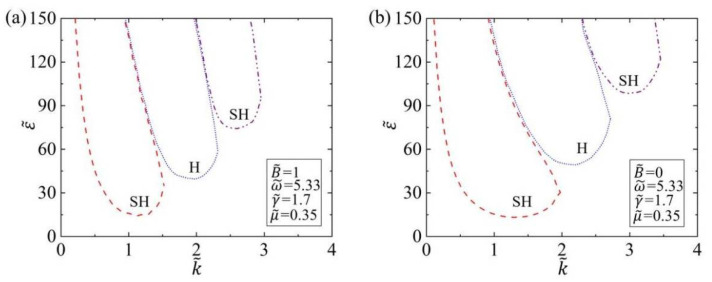
The marginal stability curves of subharmonic mode and harmonic mode for (**a**) $\tilde{B}=1$ and (**b**) $\tilde{B}=0$, respectively. The dimensionless parameters are $\tilde{\omega }=5.33$,$\tilde{\gamma }=1.7$, and $\tilde{\mu }=0.35$. The three tongue-shaped curves all have the lowest point, and Faraday instability in the viscous fluid covered with the elastic polymer film can be triggered when $\tilde{\varepsilon }$ is above the lowest point.

**Figure 3 polymers-14-02334-f003:**
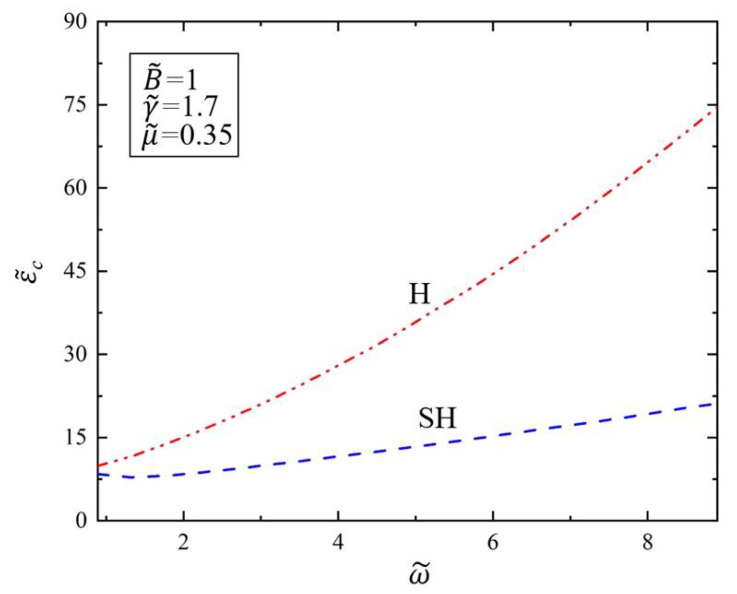
Variation of the critical amplitude ${\tilde{\varepsilon }}_{c}$ of the Faraday instability with driving frequency $\tilde{\omega }$. The dimensionless parameters are $\tilde{\gamma }=1.7$, $\tilde{\mu }=0.35$, $\tilde{B}=1$. The critical amplitude ${\tilde{\varepsilon }}_{c}$ increases as the driving frequency $\tilde{\omega }$ increases.

**Figure 4 polymers-14-02334-f004:**
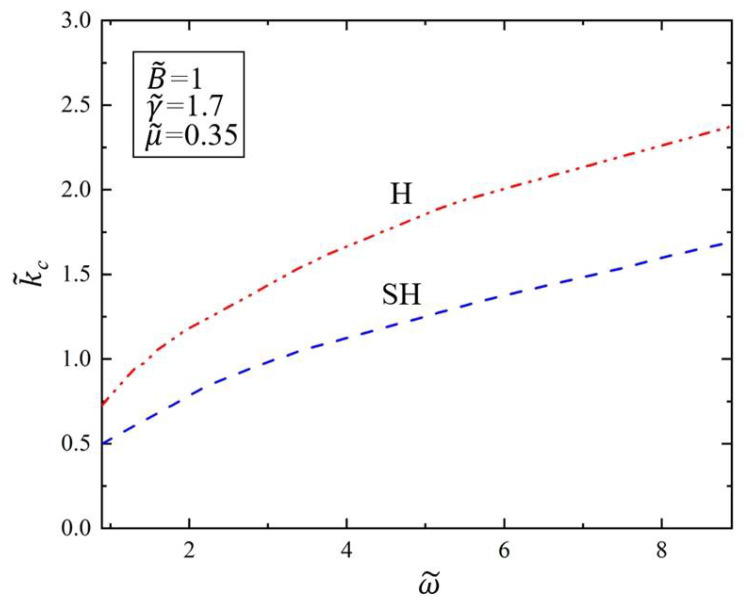
Variation of the critical wave number ${\tilde{k}}_{c}$ of the Faraday instability with driving frequency $\tilde{\omega }$. The dimensionless parameters are $\tilde{\gamma }=1.7$, $\tilde{\mu }=0.35$, and $\tilde{B}=1$. The critical wave number ${\tilde{k}}_{c}$ increases with the increase in driving frequency $\tilde{\omega }$.

**Figure 5 polymers-14-02334-f005:**
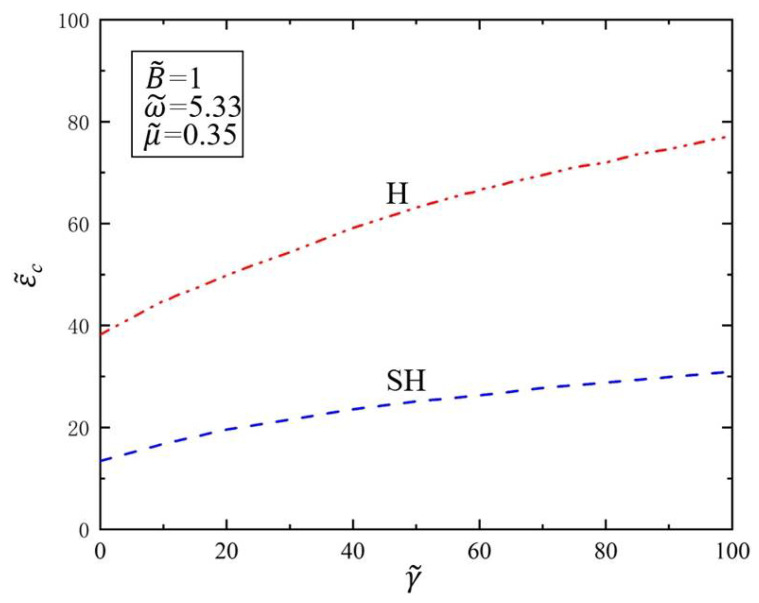
Variation of the critical amplitude ${\tilde{\varepsilon }}_{c}$ of the Faraday instability with surface tension $\tilde{\gamma }$. The dimensionless parameters are $\tilde{\omega }=5.33$, $\tilde{\mu }=0.35$, and $\tilde{B}=1$. The critical amplitude ${\tilde{\varepsilon }}_{c}$ increases with the increase in surface tension $\tilde{\gamma }$.

**Figure 6 polymers-14-02334-f006:**
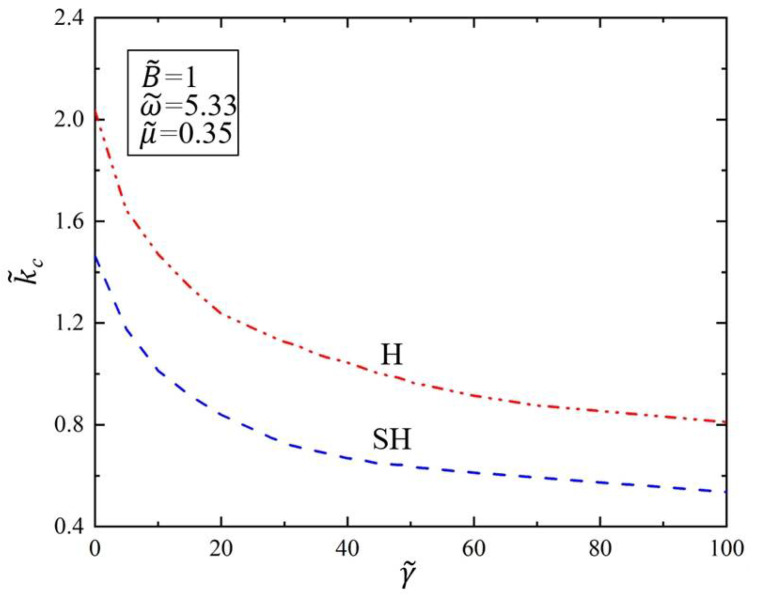
Variation of the critical wave number ${\tilde{k}}_{c}$ of Faraday instability with surface tension $\tilde{\gamma }$. The dimensionless parameters are $\tilde{\omega }=5.33$, $\tilde{\mu }=0.35$, $\tilde{B}=1$. The critical wave number ${\tilde{k}}_{c}$ decreases as the surface tension $\tilde{\gamma }$ increases.

**Figure 7 polymers-14-02334-f007:**
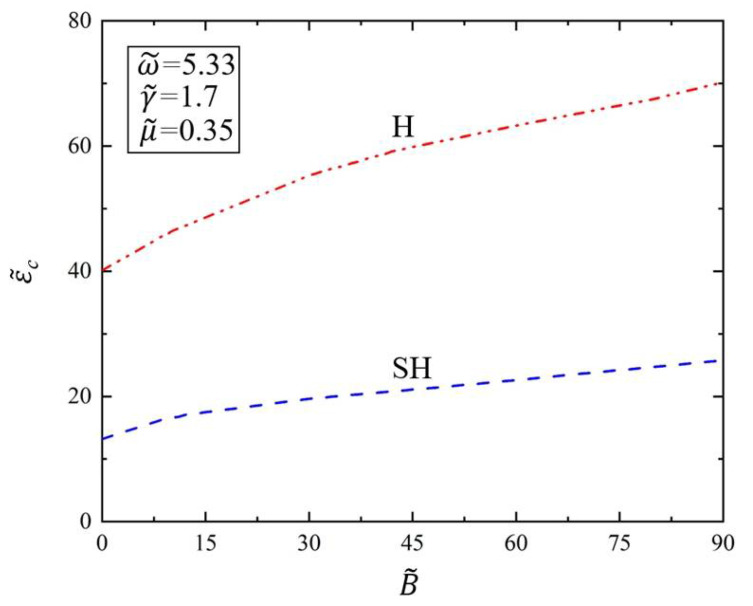
Variation of the critical amplitude ${\tilde{\varepsilon }}_{c}$ of Faraday instability with the bending stiffness $\tilde{B}$. The dimensionless parameters are set as $\tilde{\omega }=5.33$, $\tilde{\gamma }=1.7$, and $\tilde{\mu }=0.35$. The critical amplitude ${\tilde{\varepsilon }}_{c}$ increases with the increase in bending stiffness $\tilde{B}$.

**Figure 8 polymers-14-02334-f008:**
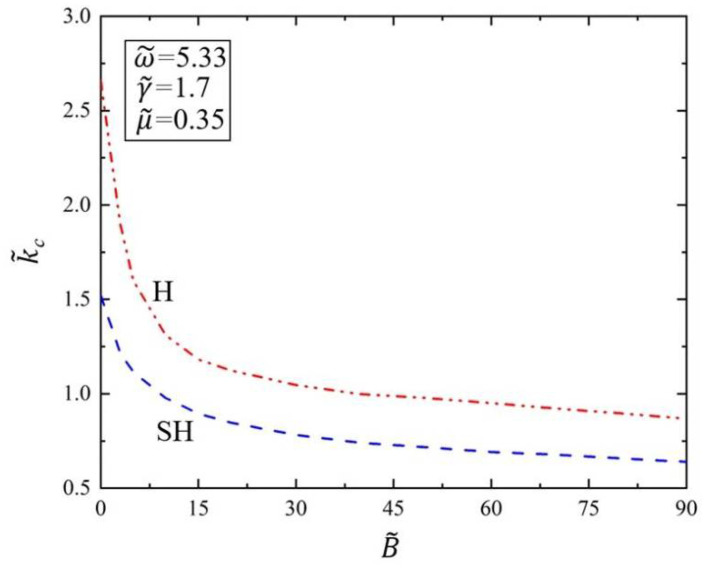
Variation of the critical wave number ${\tilde{k}}_{c}$ of Faraday instability with the bending stiffness $\tilde{B}$. The dimensionless parameters are set as $\tilde{\omega }=5.33$, $\tilde{\gamma }=1.7$, and $\tilde{\mu }=0.35$. The critical wave number ${\tilde{k}}_{c}$ decreases with the increase in bending stiffness $\tilde{B}$.

**Table 1 polymers-14-02334-t001:** Material properties and geometric parameters.

Parameter	Definition	Value	Units
H	Thickness of viscous fluid layer	2	mm
t	Thickness of the elastic polymer film	0.02	mm
γ	Surface tension	$10^{-5}$	N/m
ω	Driving frequency	10~100	Hz
μ	Kinematic viscosity	1~10	$cm^{2}/s$
B	Bending stiffness	$10^{-8}\sim 10^{-4}$	$\mathrm{kg}\times m^{2}{/s}^{2}$

**Table 2 polymers-14-02334-t002:** Dimensionless parameters.

**Parameter**	$\tilde{\gamma }$	$\tilde{\omega }$	$\tilde{\mu }$	$\tilde{B}$
**Value**	0~100	0.89~8.9	0.125~12.5	0~90

## Data Availability

The data that support the findings of this study are available upon reasonable request from the authors.

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
