# Peer review of "Faraday Instability in Viscous Fluids Covered with Elastic Polymer Films"

_polymers, 2022, doi:10.3390/polym14122334_

Round 1

Reviewer 1 Report

In this manuscript, Liu et al. have investigated the Faraday instability of an elastic polymer film covering the surface of a viscous fluid layer, based on the hydrodynamic theory. They have formulated the film stability in terms of the critical acceleration amplitude and critical wave number and the relation of both factors with surface tension and bending stiffness is discussed. The manuscript is well written and understandable and the conclusions are supported by the results. I recommend publication of this manuscript subject to the following minor revisions:

1-Page 2, introduction: The applications of Faraday instability, investigated in this article, is well discussed. A recent application is in the field of solution of Janus colloidal particles (i.e. the elastic polymer film) in a solvent (i.e. the viscous fluid); see for example J. Chem. Theory Comput. 2022, 18, 1870 and J. Chem. Theory Comput. 2021, 17, 1742. By tuning the magnitude of anisotropic colloid-solvent interactions, Faraday waves of different amplitudes (depending on the colloid-solvent interfacial tension and the surface tension of colloid) are generated.  Maybe addressing this point in the text can further illuminate the domain of application of the subject discussed in the present article.

2-In thin polymer (polymer-solvent) films on the top of a substrate, the structure of the film is influenced by the substrate. This means that the amplitude of Faraday waves depends on the structure of substrate and the magnitude of fluid-substrate and polymer-substrate interactions. I assume that the thickness of the film (H) in the present study is large enough, so that the effects of above-mentioned parameters on the Faraday wave amplitude is negligible. Please provide a measure for the film thickness (i.e. in the micron domain or so).   

3-Variation of the critical wave number and the critical amplitude with the driving frequency, the surface tension, and the bending stiffness follow the same trend for both harmonic and subharmonic modes. Please discuss this point in the text and provide and interpretation for that in the conclusions.  

4-The relation between the surface tension and the bending stiffness is known in the literature (J. Chem. Theory Comput. 2022, 18, 2597). In my opinion the calculations on pages 13-15 (Figs 7 and 8) can be interpreted in terms of the relation between the surface tension (discussed in Figs. 5 and 6) and the bending stiffness. If the authors do so, I recommend transferring parts of the calculations on pages 13-15 to SI.  

Reviewer 2 Report

This is an excellent paper, which actually only lacks one thing to be perfect: Experimental validation of the calculations. To enable also other authors to make the respective experiments, it would be nice to propose a fluid/film combination where this should work, or to sum up the relevant parameter ranges for stiffness (expressed as modulus) and interfacial tension clearly. 

Otherwise, only one text correction seems necessary: "Compared to existing methods, such as changing the viscosity, this strategy is simple to operate and cost-effective. The text of this paper is structured as follows: Firstly, we ..."
